



# Importance of Ice Nucleation and Precipitation on Climate with the Parameterization of Unified Microphysics Across Scales version 1 (PUMASv1)

Andrew Gettelman[1], Hugh Morrison[1], Trude Eidhammer[1], Kate Thayer-Calder[1], Jian Sun[1], Richard Forbes[3], Zachary McGraw[4,5], Jiang Zhu[1], Trude Storelvmo[2], and John Dennis[1]

[1]National Center for Atmospheric Research, Boulder, CO, USA
[2]Department of Meteorology, University of Oslo, Oslo, Norway
[3]European Centre for Medium Range Weather Forecasts, Reading, UK
[4]Department of Applied Physics and Applied Mathematics, Columbia University, New York, NY, USA
[5]NASA Goddard Institute for Space Studies, NY, USA

**Correspondence:** A. Gettelman (andrew@ucar.edu)

**Abstract.** Cloud microphysics is critical for weather and climate prediction. In this work, we document updates and corrections to the cloud microphysical scheme used in the Community Earth System Model (CESM) and other models. These updates include a new nomenclature for the scheme, and the ability to run the scheme on Graphics Processing Units (GPUs). The main science changes include removing an ice number limiter and associated changes to ice nucleation, adding vapor deposition onto snow, and introducing an implicit numerical treatment for sedimentation. We also detail the improvements in computational performance that can be achieved with GPU acceleration. We then show the impact of these scheme changes on (A) mean state climate, (B) cloud feedback response to warming and (C) aerosol forcing. We find that corrections are needed to the immersion freezing parameterization without a limit on ice number. We also find that the revised scheme produces less liquid and ice, but that this can be adjusted by changing the loss process for cloud liquid (autoconversion). Furthermore, there are few discernible effects of the PUMAS changes on cloud feedbacks, but some significant reductions in the magnitude of Aerosol Cloud Interactions (ACI). Small cloud feedback changes appear to be related to the implicit sedimentation scheme, with a number of factors affecting ACI.

## 1 Introduction

Cloud microphysics is critical for weather and climate. In particular, supercooled liquid water and the fraction of condensate in the form of supercooled liquid at high latitudes has been shown to be important for mean state climate (Bodas-Salcedo et al., 2019) and weather (Forbes et al., 2015) biases. Supercooled liquid water is also important for understanding cloud feedbacks and climate sensitivity (Tan et al., 2016; Gettelman et al., 2019a; Tan and Storelvmo, 2019). For example, the Community Earth System Model version 2 (CESM2, Danabasoglu et al. (2020)) has been shown to have a higher supercooled liquid fraction than observed (Gettelman et al., 2020) which has been linked to a higher climate sensitivity than previous versions (Gettelman et al., 2019a). In addition, cloud microphysics controls the formation of precipitation, which is often too light





and too frequent in large scale weather and climate models (Stephens et al., 2010). This frequency bias has been shown to be directly attributable to cloud microphysics (Gettelman et al., 2021). In this manuscript, we describe adjustments and corrections to the ice microphysics and precipitation process in an advanced cloud microphysics scheme used in several global models and how these changes can have a significant effect on simulated climate.

The state of the art for global model cloud microphysics is the use of bulk two moment schemes, such as the scheme developed by Morrison and Gettelman (2008) (MG1). The CESM2 atmosphere model uses cloud microphysics described by Gettelman and Morrison (2015) and Gettelman et al. (2015), termed 'MG2' (Morrison-Gettelman version 2). The scheme has been updated further to include rimed ice by Gettelman et al. (2019b) (MG3). Most two moment bulk schemes utilize classes of hydrometeors (such as cloud liquid, cloud ice, and snow). In such schemes, number and mass are predicted for each class

and hydrometeors are treated using a size distribution with a fixed functional form. Two moment schemes are widely used in mesoscale models (Thompson and Eidhammer, 2014) at resolutions of a few kilometers, but less commonly used in global models (e.g., Lohmann et al., 1999, 2007; Michibata et al., 2019). The scheme of Morrison and Milbrandt (2015) uses a single hydrometeor class to represent all ice particles, thereby unifying the treatment of cloud ice and snow, and was implemented in the MG1 scheme by Eidhammer et al. (2016).

Our purposes in this manuscript are several. First, we document updates and corrections to the CESM2 cloud microphysical scheme. These include a new nomenclature for the scheme, and the ability to run the scheme on Graphics Processing Units (GPUs). We detail the improvements in computational performance that can be achieved with GPU acceleration. We then show the impact of the changes and corrections on (A) mean state climate, (B) cloud feedback response to warming and (C) aerosol forcing. Note that we are not presenting a new set of tuned model configurations at this time. Rather, we show the impacts

of changes and improvements in the scheme, particularly related to ice and mixed-phase processes, with the goal of building toward a cloud physics scheme that can work across scales (as shown in Gettelman et al., 2019c).

We describe the model and microphysics scheme updates in Section 2. Section 3 describes the methodology, simulations and evaluation. Section 4 presents results, and Section 5 has discussion and conclusions.

## 2   Model Description

### 2.1   Model Description, CESM2

CESM2 (Danabasoglu et al., 2020) features the Community Atmosphere Model version 6 (CAM6) as its atmosphere model. CAM6 uses a two-moment stratiform cloud microphysics scheme MG2 (Gettelman and Morrison, 2015; Gettelman et al., 2015) with prognostic liquid, ice, rain and snow hydrometeor classes. As a two-moment bulk scheme, MG2 has prognostic variables for number and mass mixing ratios for each of the hydrometeor classes. It also permits ice supersaturation and is

described further below.

MG2 is also coupled to a 4-mode aerosol model (Liu et al., 2016) with liquid activation following Abdul-Razzak and Ghan (2002) and features natural and anthropogenic aerosols. CAM6 includes a physically based mixed phase ice nucleation scheme (Hoose et al., 2010) and accounts for preexisting ice for nucleation (Shi et al., 2015). MG2 is coupled to a unified moist





turbulence scheme, Cloud Layers Unified by Binormals (CLUBB, Golaz et al., 2002; Larson et al., 2002), implemented in
CAM by Bogenschutz et al. (2013). CLUBB treats boundary layer moist turbulence, stratiform clouds and shallow cumulus.
CAM6 uses an ensemble plume mass flux deep convection scheme (Zhang and McFarlane, 1995; Neale et al., 2008) with
simple microphysics. The radiation scheme is the Rapid Radiative Transfer Model for General Circulation Models (RRTMG)
(Iacono et al., 2000).

## 2.2 Cloud Microphysics

MG2 (Gettelman and Morrison, 2015) is an update of the original MG scheme (Morrison and Gettelman, 2008; Gettelman et al.,
2008). The original MG scheme was implemented in CAM5 as described by Gettelman et al. (2010). MG2 added prognostic
rain and snow and is used in CAM6. Many of the process rate treatments are similar to the bulk two moment microphysics
scheme of Morrison et al. (2005) developed for mesoscale models. Gettelman et al. (2019b) further updated the MG2 scheme
to 'MG3' by adding rimed ice (with a switch to represent either graupel or hail). Thus, MG3 includes two additional prognostic
variables for graupel/hail number and mass mixing ratios. In separate work, Eidhammer et al. (2016) and Zhao et al. (2017)
updated the scheme to use a single category for cloud ice and snow. This has not been implemented yet in MG2 or MG3, and
integration is a subject for future work.

Versions of MG microphysics have been adapted for use in the NASA-GEOS (Goddard Earth System Model for Earth
Observation) model (Barahona et al., 2014), The Geophysical Fluid Dynamics Laboratory (GFDL) Atmospheric Model (AM)
(Salzmann et al., 2010) and the NASA Goddard Institute for Space Studies (GISS) model (Cesana et al., 2019). The scheme
is also available in the idealized Kinematic Driver (KID) (Shipway and Hill, 2012; Gettelman et al., 2019b). MG1 and MG2
are also used in a suite of earth system models derived from CESM1 and CESM2. Analyses have been conducted against
observations for sub-grid process rate formulations (Lebsock et al., 2013) and against in-situ aircraft flights for supercooled
liquid water (Gettelman et al., 2020). There has been significant work on numerical timestepping issues detailed by Gettelman
and Morrison (2015) and Santos et al. (2020).

There are now a large number of people ontributing to community development of the MG scheme. Thus, the 'MG' name
is no longer appropriate and we are now naming the scheme the Parameterization for Unified Microphysics Across Scales
(PUMAS). This manuscript describes the updates and changes from MG3 to PUMAS (version 1).

As part of this update, we have tested and implemented several science and software changes to PUMAS beyond the MG3
scheme (Gettelman et al., 2019b). These changes are:

1. Refactoring of the ice number limiter.

2. Reverting Immersion Freezing to Bigg (1953).

3. Allowing vapor depositional growth of snow.

4. Fall speed correction for rain/snow/graupel.

5. An option for implicit sedimentation calculation.




6. An option for cloud droplet accretion by rain to act on newly autoconverted rain.

7. Addition of OpenACC (GPU accelerator) directives.

8. Optional switches to replicate processes in the European Centre for Medium Range Weather Forecasts (ECMWF) Integrated Forecast System (IFS).

### 2.2.1 Ice Number Limiter

In all versions of MG microphysics, a cloud-ice-number limiter is applied such that, when integrated over the time step, ice nucleation cannot increase the total ice concentration beyond the concentration of available ice nuclei. This approach limits ice number from all processes except sedimentation and homogeneous freezing of cloud and rain drops. The concentration of available ice nuclei in this approach, *NIMAX*, includes the Liu and Penner (2005) homogeneous *NIHOM* and heterogeneous *NIHET* cirrus ice nucleation and the empirical Meyers et al. (1992) ice nucleation as a function of temperature for the mixed phase *NIMEY*:

$$NIMAX = NIHET + NIHOM + NIMEY \qquad (1)$$

When the Hoose and Möhler (2012) Classical Nucleation Theory (CNT) based scheme was introduced in CAM6 and the Meyers et al. (1992) ice nucleation removed, the ice nucleation source from Hoose and Möhler (2012) *NICNT*, comprising immersion, deposition and contact freezing terms, was not added to *NIMAX* (the equation above was unchanged). In addition, the ice nucleation source from Meyers et al. (1992) was set to zero (NIMEY = 0). As a result *NIMAX* in the mixed phase regime (where $NICNT > 0$) is too low, preventing the model from adding ice crystals through local nucleation as intended.

The reduced *NIMAX* leads to lower ice number concentration, and hence increased size and increased fall speed and loss through sedimentation (Shaw et al., 2021) as well as affecting the balance of ice and supercooled liquid at high latitudes. In CAM6, this increases the supercooled liquid fraction when the NIMAX limiter is removed (Shaw et al., 2021). The CAM6 simulated supercooled liquid fraction is higher than observed in key regions (Gettelman et al., 2020). The supercooled liquid fraction and ice nucleation have been implicated in altering cloud feedbacks and climate sensitivity (Gettelman et al., 2019a; Tan and Storelvmo, 2019), suggesting that how the ice number limiter is applied may be important. In this update (all simulations but the 'control'), the *NIMAX* limiter on ice nucleation has been removed and instead an overall cap on in-cloud ice number concentration is added to the end of the microphysics (set at $100 \, \text{cm}^{-3}$). This is similar to the ice number concentration limiter applied to many two-moment microphysics schemes in cloud and mesoscale models (Morrison et al., 2005; Milbrandt and Yau, 2005; Thompson and Eidhammer, 2014). Zhu et al. (2022) implemented a similar change to develop a version of CESM2 for use in paleoclimate applications.

### 2.2.2 Reverting Immersion Freezing

When the ice number limiter is removed, the immersion freezing component of the classical nucleation theory (CNT) mixed phase ice nucleation scheme of Hoose et al. (2010) was found to produce an unrealistically high number of ice crystals near





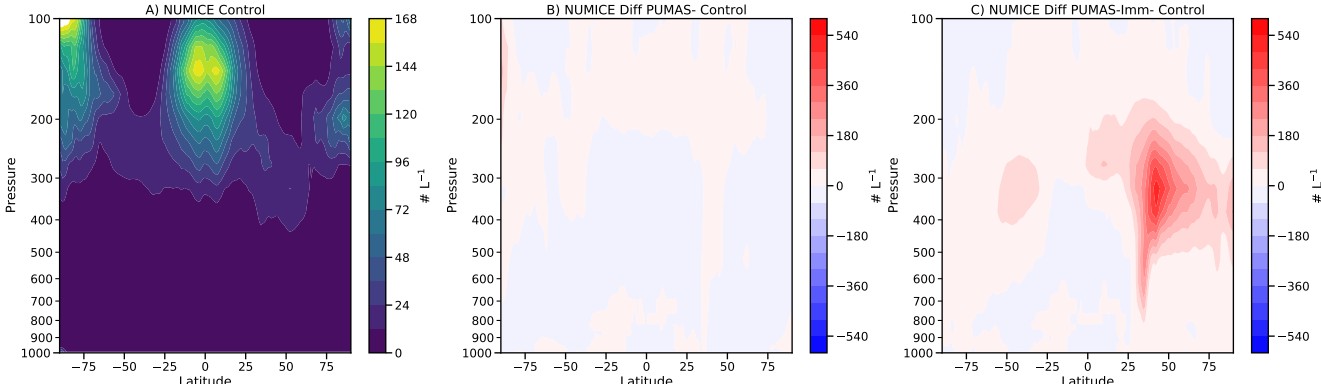

**Figure 1.** Zonal annual mean latitude height plots of A) Cloud ice number (NUMICE) from the MG3 Control simulation, B) PUMAS - Control difference, and C) PUMAS-Imm (CNT Immersion Freezing) - Control difference.

the low end of the mixed phase temperature range (-38°C), heavily concentrating ice crystals in the mid-to-upper atmosphere near desert regions (sources of mineral dust that acts as ice nucleating particles) to an extent not evident in satellite retrievals (Sourdeval et al., 2018). Figure 1C illustrates the large increase in ice number when PUMAS is implemented with CNT

immersion freezing. As a result, this component of the CNT mixed phase ice nucleation is switched off, reverting to the empirical treatment of Bigg (1953) used in MG1 which depends on temperature and cloud drop size, not the presence of aerosol particles. The use of Bigg (1953) immersion freezing does not represent the strong dependence of ice nucleation on proximity to mineral dust sources seen with the current CNT implementation. This may represent mixed-phase clouds more like in the real world, where ice at mixed-phase temperatures can additionally nucleate on aerosol species not included in CESM, such as

bioaerosols (Kanji et al., 2017). The ice number difference relative to CAM6 goes away if the Bigg (1953) parameterization for immersion freezing of cloud drops is used in PUMAS (Figure 1B). A complete explanation for the excessive ice nucleation at the edge of the applied temperature range for CNT is a subject of future work. However, one possibility is that without considering a full aerosol budget that removes cloudborne aerosols, CNT results in excessive nucleation and high ice numbers. This will influence the simulations described in Zhu et al. (2022), wherein CNT without an ice number limiter was shown to

produce improvements in paleoclimate simulations though very possibly at the expense of realistic mixed-phase clouds and associated feedbacks. The overactive CNT nucleation may also have enabled the strong sensitivity to numerical sub-stepping of the microphysics in that study's model version.

### 2.2.3  Vapor Deposition onto Snow

All versions of MG include vapor deposition onto ice in ice supersaturated conditions, but not vapor deposition onto snow. For

physical consistency, we have now added vapor deposition onto snow in PUMAS. This follows the same method as for vapor deposition onto cloud ice as described in Morrison and Gettelman (2008), equations 21-22, but uses snow size distribution





parameters instead. Grid mean temperature, humidity and snow mass mixing ratio are used in the calculation. The calculation is limited to prevent over-depletion of ice supersaturation (conditions cannot become ice sub-saturated from vapor deposition).

### 2.2.4 Precipitation Fall Speed

In the MG1, MG2, and MG3 parameterizations, sedimentation of all cloud and precipitation categories is sub-stepped to maintain numerical stability for the explicit first-order upwind calculation. Precipitation has zero fall speed if it hits a level with no precipitation during a sedimentation sub-step. This will result in "stalling" of the precipitation for the remainder of the full (not sub-stepped) time step. This can be a problem with long time steps (the standard CESM2 physics timestep is 1800s). We have implemented a correction which sets the number- and mass-weighted fall speeds for each hydrometeor category to that

at the lowest level that contains non-zero (mass mixing ratio with respect to dry air larger than $1.e^{-10}$) mass of that category before the sub-stepping and sedimentation calculation. Thus, number and mass for that category will fall at constant speeds below the lowest model level with non-zero category mass.

### 2.2.5 Implicit Sedimentation

The default method used for sedimentation in all MG versions is an explicit first-order upwind calculation. Sedimentation sub-

steps the timestep to ensure numerical stability with the fall velocity. We have added an option for a time-implicit monotonic scheme for sedimentation using a single fall speed profile. The calculation is from Guo et al. (2021) Appendix A. The mass mixing ratio $q$ of a hydrometeor at the $n+1$ time step and the $k$ vertical layer ($q_k^{n+1}$) is defined by (Guo et al., 2021, equation A3):

$$q_k^{n+1} = \frac{q_k^n \delta z_k + q_{k-1}^{n+1} v_{k-1} \delta t}{\delta z_k + v_k \delta t} \tag{2}$$

where $\delta t$ is time step, $v$ is the fall velocity, and $\delta z_k$ is the depth of the $k$-th vertical layer. Note that $k$ increases downward so the flux is dependent on what is above, and the fall speed correction will have no or little impact. The implicit scheme was originally derived from the GFDL microphysics (appendices in Harris et al. (2020) and Zhou et al. (2019)), and reduces sensitivity to time step without sub-stepping (see section 4.3). This approach is less accurate (more diffusive) at the gain of being much faster, as shown by Guo et al. (2021). In Section 4.3, we will show it does not introduce significant errors in the

solutions.

### 2.2.6 Modification of Accretion

The autoconversion parameterization in MG and PUMAS is from Khairoutdinov and Kogan (2000) (KK2000). The KK2000 autoconversion to rain tendency ($\Delta q_r = -\Delta q_c$), the change in mass mixing ratio per second, is defined as:

$$\Delta q_{rAUTO} = A q_c^B n_c^C \tag{3}$$

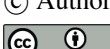



where $A$= 13.5, $B$ =2.74 and $C$=-1.1. Accretion (an additional $\Delta q_r$) is defined as:

$$\Delta q_{rACCRE} = A(q_c q_r)^B \tag{4}$$

where $A$=67 and $B$=1.15. In principle, if there is autoconversion in a long time step, autoconverted rain ($q_r$) may accrete cloud liquid. We experiment to see if this affects the aerosol-cloud interactions by allowing accretion to operate on the newly autoconverted rain, by adding it to the existing rain mass for the calcluation of accretion so it becomes:

$$\Delta q_{rACCRE} = A([q_c - \Delta q_{rAUTO}][q_r + \Delta q_{rAUTO}])^B \tag{5}$$

This allows accretion to occur during the same time step as initial rain formation from autoconversion. We hypothesize that this may increase the accretion to autoconversion ratio and that might impact the indirect aerosol forcing (Gettelman, 2015).

### 2.2.7 GPU Directives

The atmosphere model consists of many parameterizations whose calculations are grid-independent or column-independent.
Therefore, it is possible to achieve massive parallelism for a parameterization by carefully restructuring the code and increasing the number of columns (for a global model, this means increasing the horizontal resolution), which then becomes a natural fit for GPU computing (Sun et al., 2018).

In this study, the PUMAS code is GPU-enabled by using the OpenACC directives, which allows the same source code to run on either CPU or GPU and keep its maintainability. The test machines used in this paper are Cheyenne (https://www2.cisl.ucar.
edu/resources/computational-systems/cheyenne) and Casper (https://www2.cisl.ucar.edu/resources/computational-systems/casper) at NCAR. Cheyenne is a homogeneous cluster with CPU-based nodes only, while Casper is a heterogeneous cluster with the NVIDIA V100 GPUs available on some compute nodes. The performance comparison below is made between one full CPU node on Cheyenne/Casper (i.e., 36 CPU cores) and one V100 GPU on Casper. We limit the simulation length to 1 day, which we find is long enough to show the performance difference. We mainly measure the timing information of the calculations
within PUMAS and report the maximum wall-clock time among the MPI processes that is averaged over the simulation length and sub-cycles (there is by default 3 sub-steps per time step for the calculation of MG2 tendency using the 1-degree resolution). For the GPU simulation, we also measure the timing information of GPU computations and data copy between the CPU (i.e., host) and GPU (i.e., device) to better illustrate their relative computational cost. Last but not the least, we vary the maximum number of atmospheric columns per single processing element (called a 'chunk'), the CAM PCOLS parameter. As noted by
Worley and Drake (2005), there can be multiple chunks per MPI process. PCOLS is varied from 16 (the default value) to 1536 (the theoretical maximum number for 1-degree resolution and 36 MPI processes) to reveal its impact on the CPU and GPU performance.

Figure 2 shows the computational time of CPU and GPU for PUMAS without graupel. The figure compares computational times with increasing values of PCOLS. The newer Intel Skylake processors on Casper slightly outperform the previous gen-
eration Intel Broadwell processors on Cheyenne. In addition, the CPU runs consistently show longer computational time with respect to increased PCOLS, which is likely a penalty caused by lack of cache for a larger chunk size. In contrast, the GPU run





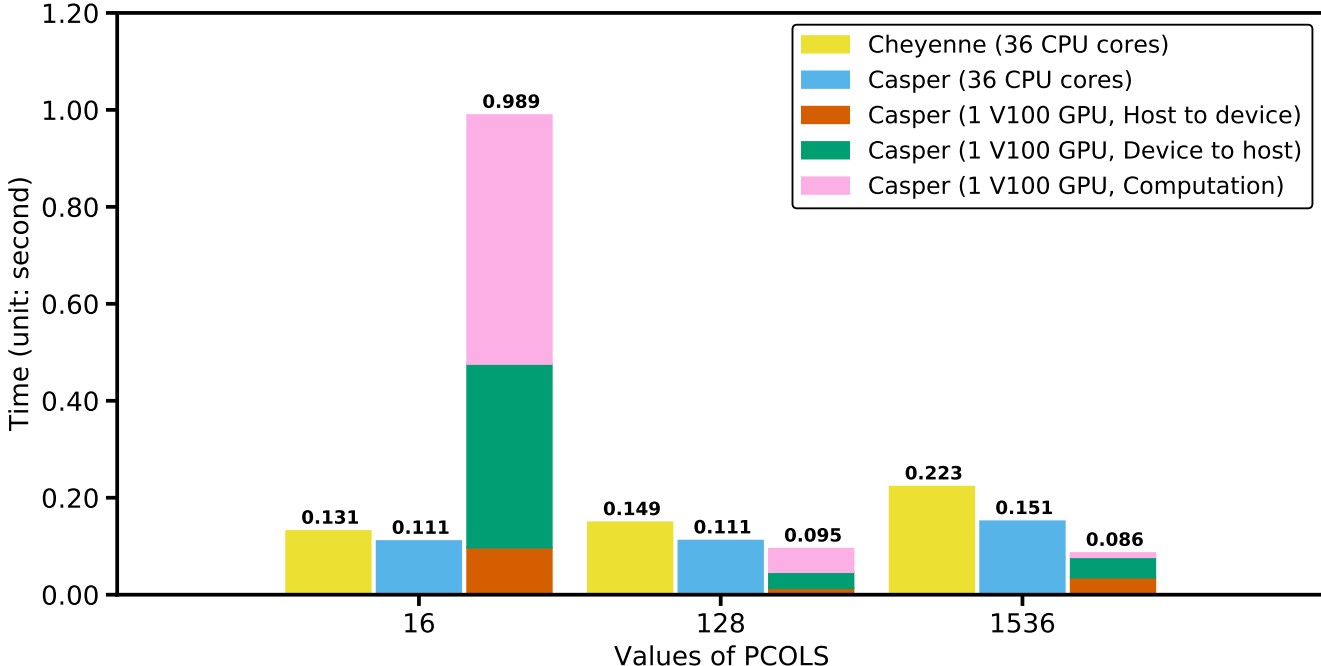

**Figure 2.** Averaged GPU-enabled MG2 run time per sub-step for different configurations on CPU and GPU. The computational time on GPUs is further divided into time spent copying data from the host to the device, from the device back to host, and time spent in pure computation. Simulations are performed on the Cheyenne and Casper computer system with details described in Section 2.2.7

is significantly slower than the CPU run when PCOLS = 16 and more than half of the time is spent on the computation (shown by pink bars). When PCOLS increases, more data is copied from CPU to GPU each time, meaning a higher parallelism and fewer total kernel launches on the GPU, as well as less frequent data movement between CPU and GPU. All of these factors

contribute to the rapid decrease of computational time for the GPU run with larger PCOLS. A factor of 2× to 3× total speedup for the GPU relative to CPU is observed when PCOLS = 1536. This suggests that we should expect to benefit from GPU computing only when our problem size (i.e., number of columns per GPU) is large enough. We also notice that the data movement could be more costly than computation for a smaller chunk size. Porting additional CAM physics parameterizations that use the same data to the GPU would likely reduce relative cost of data movement and result in even more effective speedups. Even

at PCOLS = 1536 the data transfer time is larger than the total computation time in Figure 2.

The encouraging results here serve as a proof of concept for porting PUMAS to GPU. A separate manuscript is under preparation to describe the technical details on porting PUMAS to GPU, optimizing the implementation and evaluating the CPU and GPU performance with different configurations.

The addition of OpenACC directives is included in all the PUMAS tests, but does not change scientific results as it results

in only round off level changes.





### 2.2.8 Additional Switches

In a paper in preparation (Gettelman, et al 2022, in preparation), an additional series of options were added to the PUMAS scheme to adjust the fall speed, evaporation rate of cloud liquid and sedimenting condensate, and ability to use different ice nucleation options. The changes enhance portability, and enable the PUMAS code to run in the ECMWF IFS system. This code is currently available in the PUMAS github repository, but detailed in other work.

## 3 Methodology

Our analysis looks at the impact of the changes described in section 2 on the mean state present day climate, as well as the implications for climate forcing and climate feedbacks. Cloud microphysics impacts the forcing of climate via interactions with aerosol particles in the atmosphere that change cloud condensation nuclei (CCN), as first noted by Twomey (1977). This has significant possible feedbacks on cloud drops and radiation, as recently reviewed by Bellouin et al. (2020). Cloud microphysical responses to increased CCN and ice nucleus (IN) concentrations are a critical forcing agent for climate. Furthermore, clouds respond to the environmental state when the planet warms due to other factors (such as anthropogenic greenhouse gases). These cloud feedbacks are the largest uncertainty in determining the climate sensitivity – the response of the climate system to an imposed forcing (e.g., Gettelman et al., 2016; Sherwood et al., 2020). We assess how both aerosol forcing and climate feedbacks change due to changes in the cloud microphysics with detailed simulations and analysis.

### 3.1 Simulations

We conduct several 6 year long simulations using near present day cyclic boundary conditions for the year 2000. This includes averaged levels of greenhouse gases and atmospheric oxidants, and emissions of aerosols for the period 1995-2005. We also use averaged Sea Surface Temperatures (SSTs) from observations (Hurrell et al., 2010). Data are compiled monthly and smoothly interpolated between months. Model resolution for CAM6 is 0.9° lat x 1.25° longitude with 32 levels in the vertical up to 10 hPa and a standard physics timestep of 1800s, and 3 coupling periods between microphysics and cloud 'macrophysics' (turbulence and large scale condensation), for a default microphysics timestep of 600s. Six years is sufficient that the differences between past, present and future configurations (see below) are much large than the inter-annual standard deviation of global quantities (e.g. the signals are larger than internal variability).

For each of the cases described below in Section 4.1 we conduct the following experiments. The first is a simulation with Present Day (PD, year 2000) boundary conditions including SSTs, Greenhouse Gases (GHGs) and aerosol emissions. Next, we conduct simulations with (A) 1850 estimated aerosol emissions, termed 'Pre-Industrial' (PI) and (B) SST increased uniformly by +4K (SST4K). All other boundary conditions (especially GHGs and SSTs) are the same. When compared with the PD simulation, PI assesses the impact of aerosol forcing, and SST+4K assesses the impact of cloud feedbacks, being representative of the full model response to warming as first noted by Cess et al. (1989), and verified for CESM2 by Gettelman et al. (2019a).





For detailed analysis, we also conduct single column model simulations with the Single Column Atmosphere Model (SCAM) that is available as a part of CAM (Gettelman et al., 2019c). SCAM runs exactly the same atmospheric physics parameterization code as the full 3D simulations. We focus on the Atmospheric Radiation Measurement (ARM) program June 1997 case for a month over the central great plains (Oklahoma, USA) called the 'ARM97' case, as well as the Mixed Phase Clouds Experiment

(MPACE) case from October 2004 over the North Slope of Alaska. Cases are detailed in Gettelman et al. (2019c). The standard timestep in SCAM is 1200s, with 4 microphysics sub-steps for a microphysics timestep of 300s.

## 4  Results

### 4.1  Description of Sensitivity Tests

Sensitivity tests are grouped sequentially as indicated in Table 1, testing many of the items listed in Section 2.2, and with the

item numbers listed in that section. The first is a control case, using the standard CAM6 and the MG3 microphysics. All of these tests include graupel. Beyond the control case, we test all the 'PUMAS' modifications listed in section 2.2 except #8 (optional switches to replicate processes in the ECMWF IFS). An additional modification is to increase the specified diameter of ice crystals detrained from deep convection from 25 to 60 $\mu$m ('Control-Di60'). We test the impact of this change in a separate simulation based on the control case. The PUMAS simulation where we do not use the Bigg (1953) immersion freezing, and

instead use the CAM6 CNT version (#2), is illustrated in Figure 1 ('PUMAS-Imm'). We then look at the impact of removing vapor deposition onto snow (#3) in PUMAS as described in Section 2.2 ('PUMAS-VapDepSn'). Next, we test removing the fall speed modifications (#4) and implicit sedimentation (#5) in PUMAS simulations in the 'PUMAS-Implicit' simulations. Because the fall speed modifications mainly impact the explicit rather than implicit treatment of sedimentation, we also tested the fall speed modifications alone with the explicit sedimentation in PUMAS (e.g., testing explicit sedimentation with the fall

speed correction). The results are basically the same as explicit sedimentation without the fall speed correction in PUMAS-Implicit (not shown), indicating that the fall speed correction has little impact. Finally, we also explore a simulation where we adjust ('tune') uncertain microphysical parameters to increase cloud water and ice to get close to the control case ('PUMAS-Tune'). This tuning is accomplished by scaling the autoconversion of cloud to rain by a further factor of 0.5 from the control case and increasing the size threshold for ice autoconversion to 1000 $\mu$m (from the control value of 500 $\mu$m).

### 4.2  Process Rates

To look in detail at the effects of changes to the microphysics scheme, we focus on single column (SCAM) simulations for the ARM97 summer case. Figure 3 illustrates Liquid (A-C), Ice (D-F), Rain (G-I) and Snow (J-L) process rates. Graupel was included in these simulations but is not shown. The left column of Figure 3 (A,D,G,J) is the control simulation, the center column (B,E,H,K) is the 'PUMAS' modified code (changes #1–7 from Section 2.2) and the right column (C,F,I,L) the

difference between the two. Differences in liquid at high altitudes mainly come from changes in ice and mixed-phase phase processes (vapor deposition). In particular there is less accretion of liquid onto snow and vapor deposition onto snow (Berg

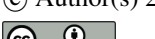



**Table 1.** Description of the simulations. Numbers in the options column correspond to the changes listed in Section 2.2.

| Name | Options | Description |
|---|---|---|
| Control | None | Baseline CAM6 with MG3 |
| Control-Di60 | None | CAM6-MG3 with detrained ice size = 60 $\mu$m |
| PUMAS | #1-7 | New Ice Limit + Vap Dep Snow + Fall speed + Accretion + Bigg (1953) Imm Frz |
| PUMAS-Imm | #1,3-7 | PUMAS with the CAM6 CNT scheme |
| PUMAS-VapDepSn | #1-2,4-7 | PUMAS without vapor deposition onto snow |
| PUMAS-Implicit | #1-3,6-7 | PUMAS without sedimentation and fall speed modifications |
| PUMAS-Tune | #1-7 | PUMAS + tuning to improve energy balance |

Snow), reducing the liquid loss rates at higher altitudes. At lower altitudes (pressures > 600 hPa) there are small changes to liquid, with less sedimentation and more loss from accretion of cloud by rain.

Ice microphysics sees larger changes, with a decrease in ice mass loss from sedimentation (leading to a positive net change). This is caused by larger ice number from removing the NIMAX limiter, and results in more ice (and hence an increase in the melting source of liquid). Cloud ice remaining longer in the atmosphere increases the availability for vapor deposition as well.

Precipitation changes in Figure 3 are also significant. Rain has a negative sedimentation tendency all the way to the surface in the revised code (Figure 3H), with reduced fall speeds higher up. Snow has a positive sedimentation tendency from 800–600hPa as snow falling from above enters a layer below. In PUMAS this continues all the way to the surface (Figure 3K) because the nonzero fallspeed was added below the lowest level of precipitation. The implicit sedimentation does not seem to introduce any numerical issues with rain or snow.

Graupel changes (not shown) are modest, and come mostly from differences in the graupel collecting snow term, which is larger in PUMAS (Figure 3L).

We have explored in detail how the different parts of the PUMAS code create these differences by plotting some of the changes separately (not shown). The upper level increases in cloud ice deposition and sublimation come from the ice number limiter change. The limiter also reduces low level accretion of cloud liquid ($q_c$) to rain (q$_r$) and snow (q$_s$). The implicit sedimentation results in more accretion of cloud liquid. The new accretion increases the accretion of cloud liquid as well, but reduces cloud ice deposition/sublimation. Thus, not all the changes act in the same way. The changes due to accretion including newly autoconverted rain have a smaller impact (limited to increasing low level accretion) than the explicit sedimentation or ice limiter changes.

## 4.3 Timestep Sensitivity

We have also used the SCAM cases as a way to explore the time step sensitivity of the schemes. To focus on the microphysics, we set a 1200 s physics timestep in SCAM, with 4 sub-steps for the CLUBB unified turbulence scheme. We then can sub-step the microphysics within that 300 s sub-step. This takes the same condensation from turbulence and passes it to the microphysics, which we run from 300 s to 10 s (e.g., 1 to 30 microphysics-sub steps). The results are illustrated in Figure 4 for the June 1997





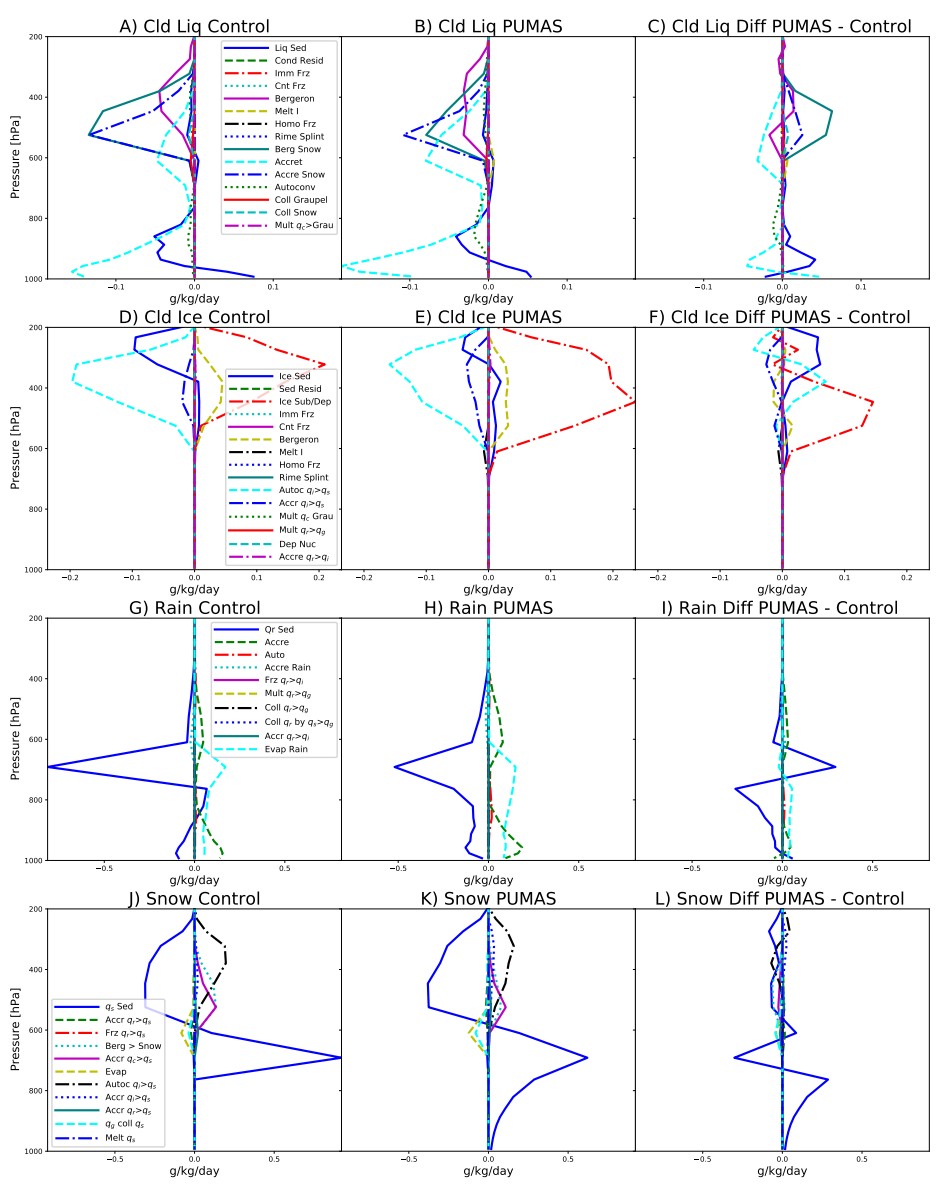

**Figure 3.** Mass mixing ratio process rate profiles from single column model (SCAM) simulations. Left column (A,D,G,J) is the MG3 Control simulation, the center column (B,E,H,K) the PUMAS code (Table 1, changes #1–7) and the right column (C,F,I,L) the difference. Top row (A,B,C) shows results for cloud liquid, then cloud ice (D,E,F), rain (G,H,I) and snow (J,K,L).



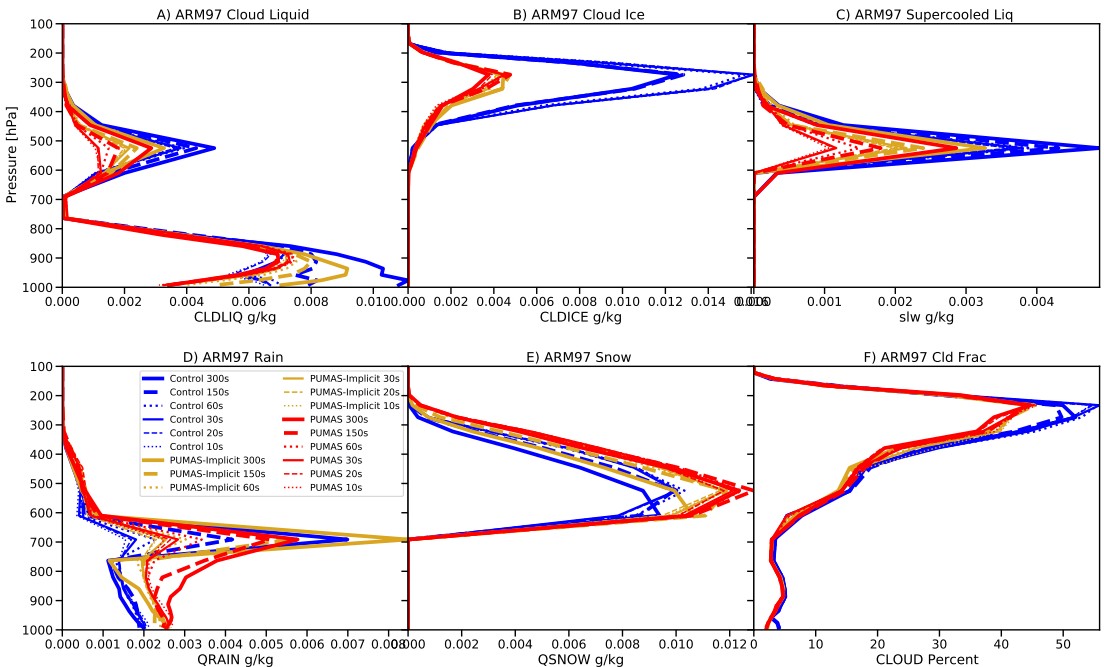

**Figure 4.** Time averaged single column model (SCAM) profiles from the ARM97 case with different time steps as indicated in the legend. Blue: MG3 Control cases, Orange: PUMAS-Implicit, Red: PUMAS. Simulation names are defined in Table 1. Results are shown for A) cloud liquid, B) cloud ice, C) supercooled liquid, D) rain, and E) snow mass mixing ratios and F) cloud fraction. Thicker lines indicate longer timesteps.

ARM97 case. We focus on the control case (MG3), all the PUMAS modifications (PUMAS) and the PUMAS without the implicit sedimentation and fall speed corrections (PUMAS-Implicit). Other sensitivity cases are similar to PUMAS. Figure 5 illustrates the same tests but for the Mixed Phase Clouds Experiment (MPACE) in October 2004 over the North Slope of Alaska.

Several results stand out. First, cloud liquid and rain (Figure 4 A and D) are fairly sensitive to time step changes for the SCAM ARM97 case in the control (MG3) configuration. The same is true for supercooled liquid (Figure 4C). The PUMAS code reduces the timestep sensitivity of most of the hydrometeors. This is particularly true for the MPACE case Figure 5, where cloud and supercooled liquid, as well as cloud ice has much less variability across timesteps than in the control MG3 case. Most of this improvement is not seen if the fall speed and sedimentation changes are removed, indicating that implicit sedimentation

contributes substantially to the reduced sensitivity of solutions to timestep (as expected). Figures 4B and C indicate reductions in supercooled liquid water and ice with the microphysical changes in PUMAS.




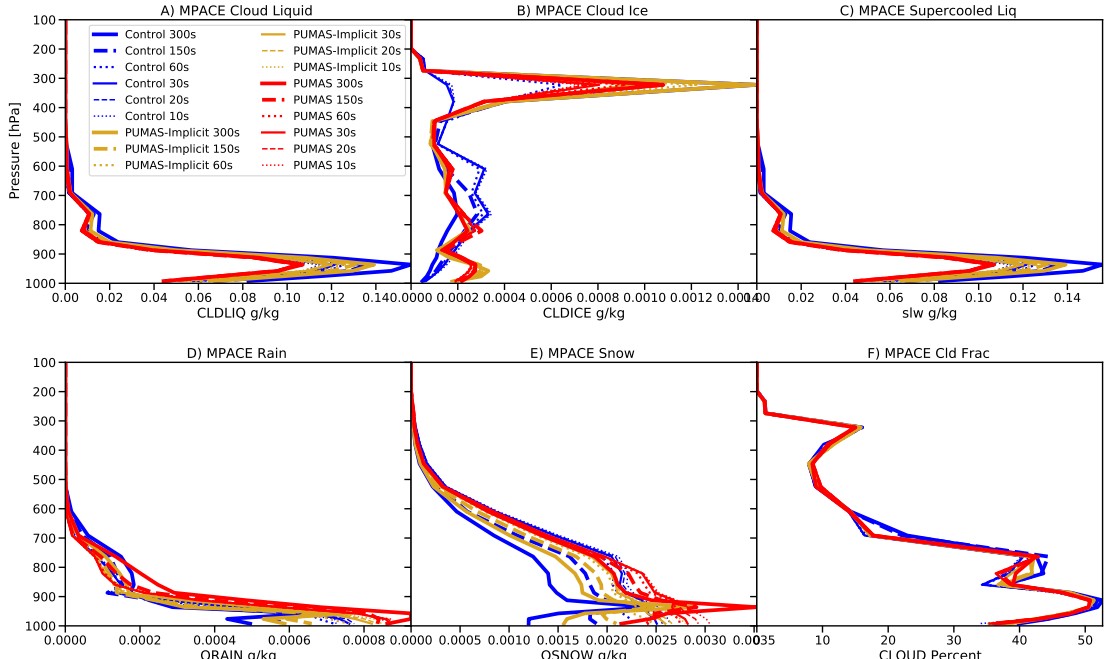

**Figure 5.** As in Figure 4 except for the MPACE case.

The MPACE case (Figure 5) has less sensitivity to time step than ARM97 for the control MG3 case, and even less sensitivity for PUMAS. Cloud liquid (all supercooled, Figure 5A and C) has little sensitivity in PUMAS, less than with MG3 or PUMAS-Implicit, again indicating the importance of implicit sedimentation. There is less cloud ice (Figure 5B) but more snow in the PUMAS simulations compared to MG3. Overall, besides the reduced time step sensitivity, PUMAS and PUMAS-Implicit give similar results for these SCAM cases.

## 4.4 Mean State Climate

Figure 6 illustrates zonal mean climatologies from 8 year global simulations of CAM6 with control microphysics (MG3) and updated PUMAS code. We test the code following the experiments in Table 1. PUMAS (Blue) is the full set of modifications. PUMAS-VapDepSn (Green) removes vapor deposition onto snow, PUMAS-Implicit (Orange) removes the fall speed changes and uses an explicit rather than implicit sedimentation calculation, and PUMAS-Tune (Red) increases liquid and ice mass with parameter changes. The full PUMAS ice modifications (Blue) reduce the overall Cloud Radiative Effect (CRE) magnitude (less negative Shortwave [SW] and less positive Longwave [LW]) from the control MG3 case by about 2.5 $\mathrm{Wm}^{-2}$ (Figure 6 G and H). This is due to lower LWP (Figure 6A) and IWP (Figure 6B). Cloud fraction, cloud top liquid number and precipitation



(Figure 6C,D and F) are similar across all simulations. The PUMA-Tune case is designed to increase liquid and ice (Figure 6A and B) to increase the magnitude of the cloud radiative effects so they are more similar to the control (Figure 6G and H), but it also decreases ice number (Figure 6E). The resulting cloud radiative effects (Figure 6G and H) are of lower magnitude than CERES LW observations, but generally within the variability of the observations (purple shading). Note the change to detrained ice size (included in all PUMAS simulations) is tested on it's own with the control model (Control-Di60), and it

increases LWP (Figure 6A) and decreases IWP (Figure 6B), likely through faster sedimentation of larger ice crystals.

Figure 7 illustrates ice fraction in the Control and PUMAS simulations. Ice fraction is calculated at each location and timestep as the ice mass mixing ratio divided by the total cloud water (ice and liquid) mass mixing ratio. An increase in ice fraction reduces the CRE from supercooled liquid water but increases CRE from ice. Cloud ice provides a negative cloud phase feedback when it is reduced under climate warming. The detrained ice size change does not impact the ice fraction in

the control simulation (Figure 7B). PUMAS features an increase in ice fraction in the tropical middle troposphere (600–300 hPa) relative to the MG3 Control simulation (Figure 7C). It appears that the implicit sedimentation change is the reason, as removing the implicit sedimentation (Figure 7E) produces ice fractions that resemble the control case. We tested that adding the fall speed correction to explicit sedimentation does not matter (not shown). The sedimentation likely impacts the evolution of ice near the freezing level. There are also some impacts at high latitudes. The tuning of clouds to increase liquid and ice

water path modifies the ice fraction due to substantial differences in the liquid and ice water path (Figure 6), generally resulting in more moderate changes to ice fraction relative to the control than with the standard PUMAS (with less condensate mass, Figure 7C).

## 4.5 Cloud Feedback

Cloud feedback is determined using a kernel adjusted cloud radiative forcing, as described by Soden et al. (2008), and imple-

mented in CESM by Gettelman et al. (2012) (also used by Gettelman et al., 2019a).

Figure 8 illustrates the kernel adjusted cloud feedback estimates, weighted by the cosine of latitude (so that the integral under the curve is the contribution to the global mean). Table 2 presents the global averages. PUMAS slightly increases the cloud feedback, lowering the LW and increasing the SW. Some of this results from the change to the detrained ice size (Control-Di60), not PUMAS itself. The PUMAS-Implicit SW feedbacks are very close to the Control-Di60 case, indicating

that differences in SW feedbacks are perhaps due to the implicit sedimentation. The change to the fall speed was tested with explicit sedimentation and does not impact feedbacks. The mechanism might be through the changes to the ice phase seen in Figure 7. Net cloud feedback (Figure 8C) is reduced from 30–70° latitude in both hemispheres, but offset with changes at higher latitudes. The PUMAS-Tune simulation with enhanced cloud water has larger cloud feedbacks, increased mostly in the SW and in the Southern Hemisphere mid-latitudes.

We have examined maps of cloud feedback by regime for the LW and SW feedbacks, and there do not appear to be large and significant systematic differences in cloud feedback between the control and PUMAS simulations. On the whole, the impact of PUMAS on cloud feedbacks is moderate and not significant.



**Figure 6.** Zonal mean annual averages from 8 year global simulations for the MG3 Control (Black), PUMAS (Blue), PUMAS-VapDepSn (Green), PUMAS-Implicit (Orange) and PUMAS-Tune (Red). Simulation names are defined in Table 1. Also shown is the CERES EBAF climatology (purple) with 1 standard deviation of annual means: A) liquid water path (LWP), B) ice water path (IWP), C) cloud fraction, D) cloud top liquid number, E) cloud top drop number, F) total precipitation, G) shortwave (SW) cloud radiatve effect (CRE), and H) longwave (LW) CRE.

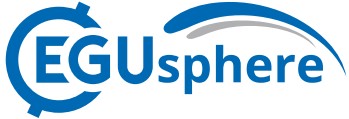

**Figure 7.** Zonal annual mean ice fraction for A) Control (MG3) and differences in ice fraction for B) Control-Di60 - Control, C) PUMAS - Control, D) PUMAS-VapDepSn - Control, E) PUMAS-Implicit - Control, and F) PUMAS Tune - Control. Simulation names are defined in Table 1.





**Table 2.** Global average kernel adjusted Cloud Feedback (CF) in W m$^{-2}$ K$^{-1}$.

| Run | Net CF | LW CF | SW CF |
|---|---|---|---|
| Control | 0.59 | 0.15 | 0.44 |
| Control-Di60 | 0.65 | 0.19 | 0.46 |
| PUMAS | 0.62 | 0.10 | 0.52 |
| PUMAS-VapDepSn | 0.64 | 0.09 | 0.54 |
| PUMAS-Implicit | 0.56 | 0.11 | 0.46 |
| PUMAS-Tune | 0.67 | 0.06 | 0.61 |

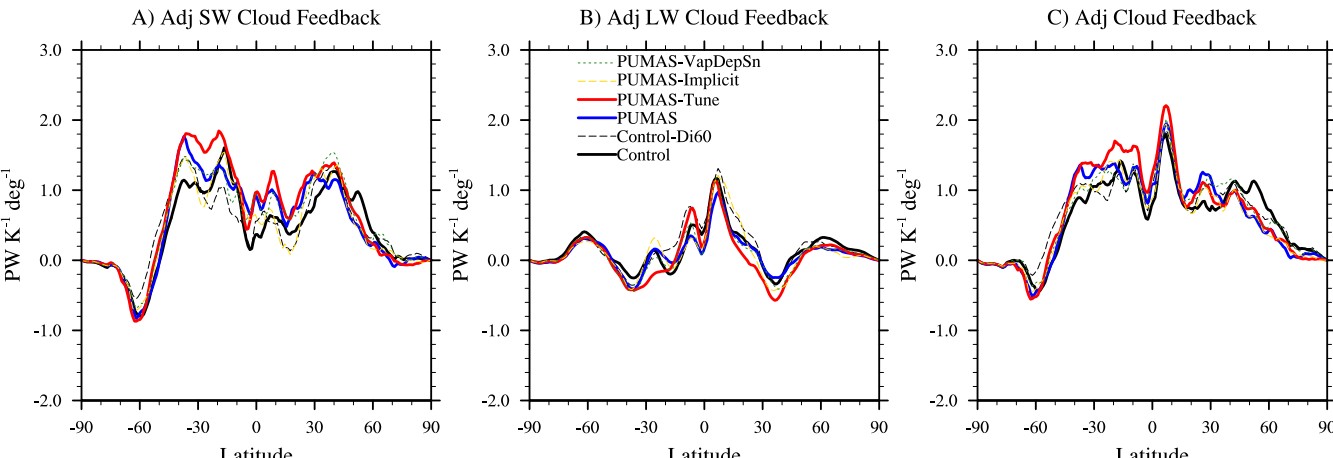

**Figure 8.** Zonal mean cloud feedback from SST+4K simulations: Control (Black), Control-Di60 (Black dashed) PUMAS (blue), PUMAS-Tune (red), PUMAS-VapDepSn (green dotted) and PUMAS-Implicit (orange dash). Simulation names are defined in Table 1. Feedbacks use kernel adjusted cloud radiative effects as described in Gettelman et al. (2013) and Gettelman et al. (2019a). Results are shown for A) SW Cloud Feedback, B) LW Cloud Feedback, and C) Net (LW+SW) Cloud Feedback.





**Table 3.** Global average forcing change due to aerosol: PD-PI (W m$^{-2}$).

| Run | $\Delta$LWP | $\Delta$IWP | $\Delta$CLD | $\Delta$CDNC | $\Delta$SWCRE | $\Delta$LWCRE | $\Delta$SW$_{clr}$ | $\Delta$RESTOM |
|---|---|---|---|---|---|---|---|---|
| Units | % | g m$^{-2}$ | % | % | Wm$^{-2}$ | Wm$^{-2}$ | Wm$^{-2}$ | Wm$^{-2}$ |
| Control | 6.4 | 0.19 | 0.60 | 41. | -2.3 | 0.53 | -0.35 | -1.9 |
| Control-Di60 | 5.8 | 0.21 | 0.40 | 40. | -2.2 | 0.63 | -0.31 | -1.8 |
| PUMAS | 6.2 | -0.65 | 0.40 | 41. | -1.9 | 0.22 | -0.17 | -1.8 |
| PUMAS-VapDepSn | 6.6 | -0.67 | 0.28 | 41. | -2.1 | 0.37 | -0.19 | -1.9 |
| PUMAS-Implicit | 6.9 | -0.87 | 0.41 | 41. | -2.1 | 0.15 | -0.21 | -2.0 |
| PUMAS-Tune | 6.0 | -3.08 | 0.28 | 36. | -1.5 | -0.23 | -0.23 | -1.9 |

## 4.6 Aerosol Forcing

Finally, we assess the impact of the changes in cloud microphysics on the aerosol forcing of climate by running 6 year sim-
ulations identical to the present day (PD) simulations, but with aerosol emissions set to 1850 emissions: 'Pre-Industrial' (PI).
The difference PD–PI represents the impact of aerosols on climate. The indirect effect of aerosols on clouds is illustrated by
the change in Cloud Radiative Effect ($\Delta$CRE). For liquid clouds this is mostly in the SW ($\Delta$SWCRE), and for ice clouds both
the SW and LW ($\Delta$LWCRE). The direct radiative effect of aerosols is the clearsky shortwave flux change ($\Delta$SW$_{clr}$). The total
effect is the all sky Residual Top Of Model ($\Delta$RESTOM) flux change. Changes in column drop number ($\Delta$CDNC) and Liquid
Water Path ($\Delta$LWP) and Ice Water Path ($\Delta$IWP) are also assessed, which drive changes in CRE for clouds with liquid. For all
quantities, the PD–PI differences are larger than twice the interannual standard deviation of global means of these quantities.

Overall, the net aerosol forcing in present day (PD-PI) defined by $\Delta$RESTOM drops by less than 10% with the PUMAS
modifications (Table 3). This occurs due to significant (50%) reductions in the negative SW clearsky forcing (direct effects of
aerosols) and 20% reductions in SW cloud radiative effect changes, but is partially offset by 50% reductions in the positive LW
forcing as well. Thus, the magnitude of the components is reduced. The PD-PI percent change in LWP is similar, but IWP goes
from an increase to a small decrease. The percent change in LWP is about 6% (increase in present) and drop number increases
by 40%, while cloud fraction increases by less than 1%. The simulations are similar for the different sensitivity tests, but this
masks regional differences.

Figure 9 illustrates the zonal mean changes in key quantities between pairs of PD and PI simulations. The gross aerosol
forcing of climate in the SW (Figure 9A) and LW (Figure 9B) is significantly reduced (closer to zero) in all of the PUMAS
simulations compared to the control: SW forcing is negative and gets less so in the N. Hemisphere. LW forcing goes from
positive in the control simulation to near zero in all three PUMAS configurations, and negative in the PUMAS-Tune simulation,
possibly due to decreases in IWP (Figure 9E). The residual net change in top of model (TOM) flux, Figure 9C, has a lower
magnitude (less negative) than the control in N. Hemisphere high latitudes for the PUMAS simulations (Figure 9C). There is a
significant reduction in the LWP and column cloud drop number (CDNUMC) response from 25–90°N latitude (Figure 9D and
F) which drives the reductions in the magnitude of the SW CRE changes (Figure 9A).



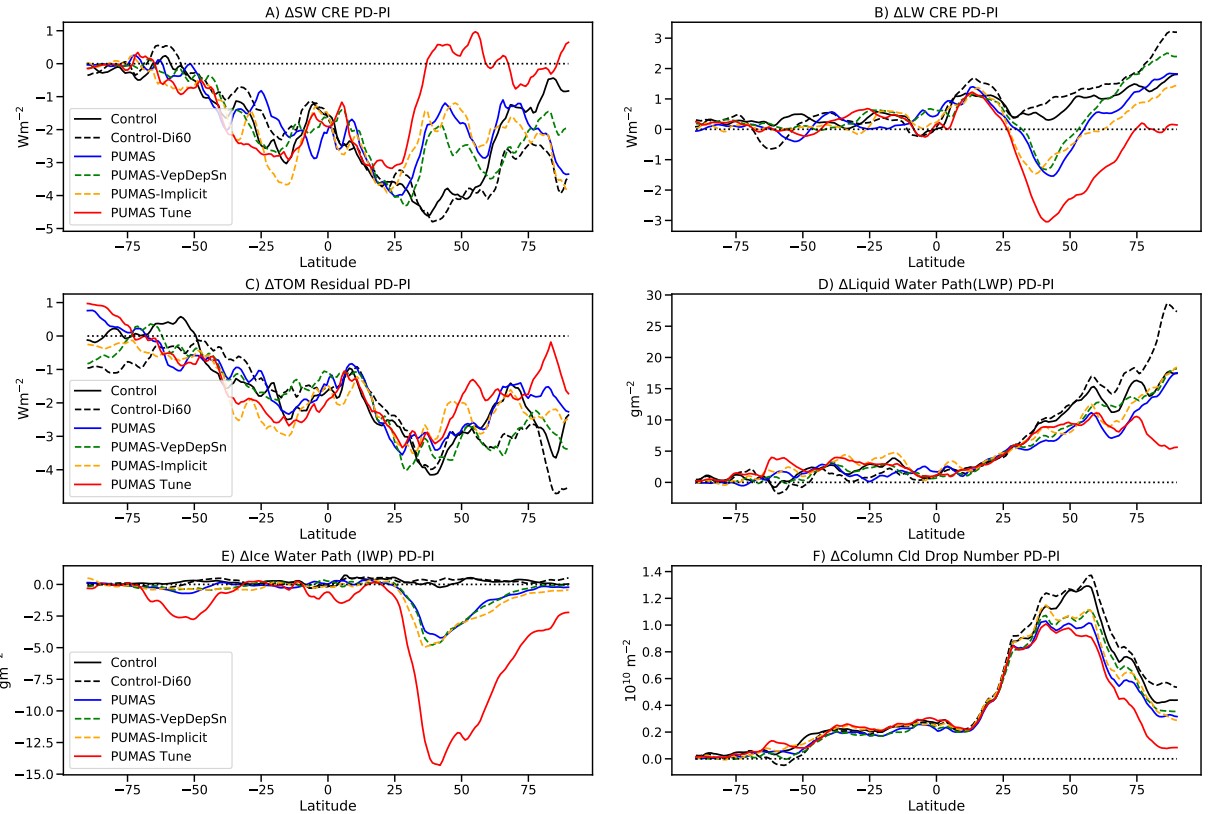

**Figure 9.** Zonal mean differences between control (PD) and 1850 aerosol (PI) simulations for 4 pairs of runs: Control (Black), Control-Di60 (Black dash), PUMAS (blue), PUMAS-VapDepSn (green dash), PUMAS-Implicit (orange dash) and PUMAS Tune (red). Simulation names are defined in Table 1. Results are shown for A) change in SWCRE, B) change in LWCRE, C) change in net TOA flux (RESTOM), D) change in LWP, E) change in IWP, and F) change in column drop number.

Since the forcing response of most of the PUMAS simulations are similar, Figure 10 compares maps of the patterns of differences (PD-PI) for the Control (left column) and PUMAS (right column) simulations. Maps of the pattern of differences in Figure 10 show clearly that these changes occur in most places across the globe. SW CRE changes (Figure 10A and B) are
reduced with PUMAS throughout the N. Hemisphere, and even into the S. Hemisphere. They are positive over North East Asia. Changes are similar over the Pacific, with large impacts in the Eastern Pacific that is a common feature of CAM ACI effects. The LW CRE changes (Figure 10 C and D) have more negative values (Figure 10D) than the Control simulation (Figure 10C), especially over East Asia, corresponding to the SW changes. Further analysis indicates this is due to a large decrease in IWP in this region in the PUMAS simulations (indicated in Figure 9E from 30–60°N). However, N.E. Asia has little impact on the
TOM residual changes as a result of the offsetting effects (Figure 10 E and F). The LWP changes (Figure 10G and H) are



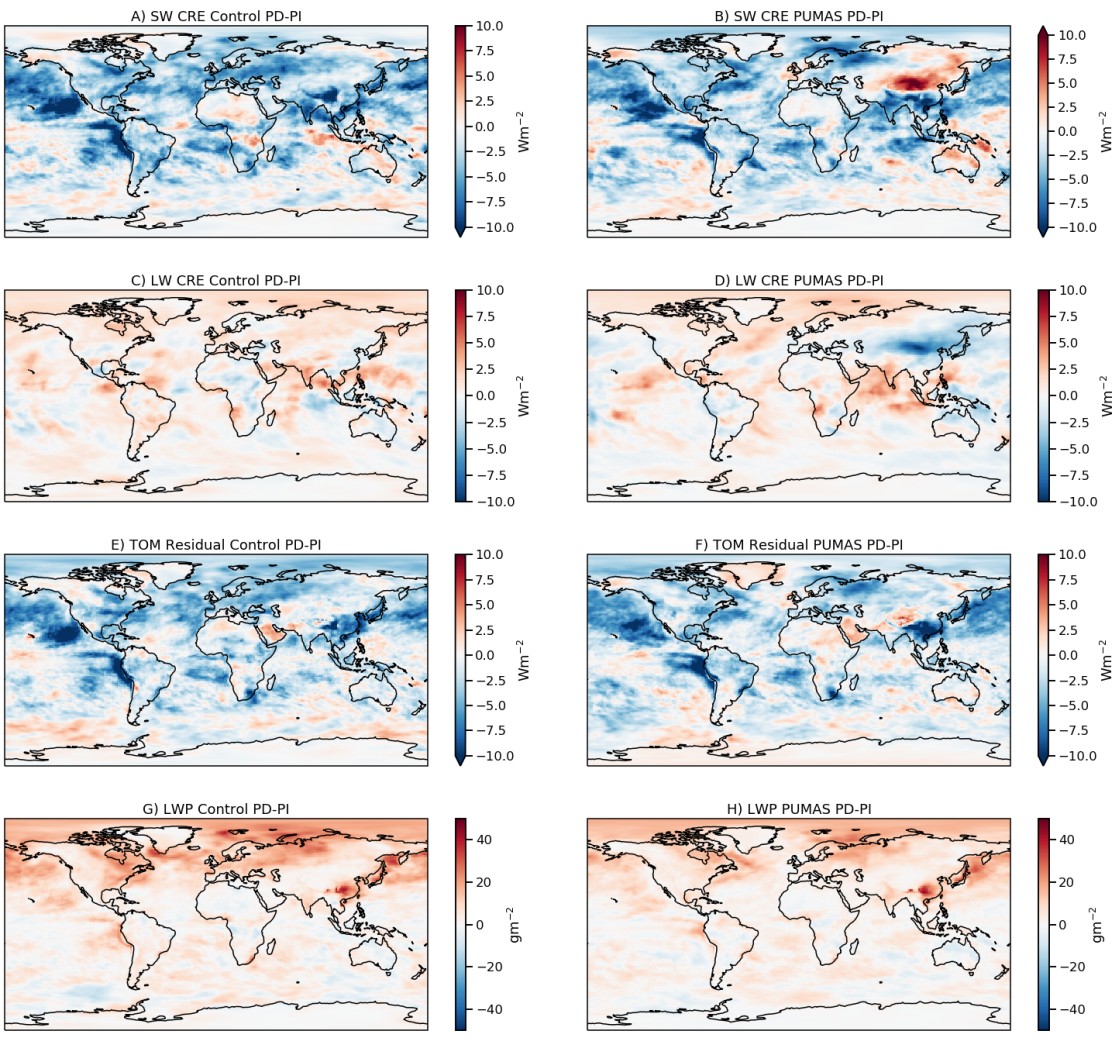

**Figure 10.** Differences between present day (PD) and 1850 aerosol (PI) simulations for Control (Left-A,C,E,G) and PUMAS+Accre (Right-B,D,F,H). Variables shown are A,B) SW Cloud Radiative Effect (SW CRE), C,D) LW Cloud Radiative Effect (LW CRE), E,F) Top of Model (TOM) Residual energy balance, and G,H) Liquid Water Path (LWP).

smaller with PUMAS and localized closer to the source regions compared to the Control. This is true especially in the Arctic (Figure 10H).



## 5   Discussion and Summary

We have described several adjustments to the MG3 cloud microphysical scheme now renamed PUMAS, including removal of an ice number limiter, adjustments to the fall speed calculation and use of an implicit numerical method for sedimentation, the addition of a process (vapor deposition onto snow), and the adjustment of accretion and the immersion freezing calculation.

The vapor deposition onto snow process has no significant effect on the simulations. Neither does the fall speed correction, even with explicit sedimentation (we do not expect much impact with implicit sedimentation). The major impact on the simulations comes from both the removal of the ice number limiter (NIMAX) and the implicit sedimentation.

The following is a summary of the main results:

1. GPU enabled microphysics code is significantly faster than the CPU version if a large number of columns can be loaded onto GPU accelerator chips. This results in a $2\times$ to $3\times$ speedup, with even more possibility for speed up at higher column counts. This would facilitate running at higher horizontal resolution.

2. Removal of the ice number limiter (change 1) results in extremely high ice numbers due to the CNT formulation of Immersion Freezing in CAM6. This significant increase in ice number is anomalous and creates a different climate state. This necessitated switching back to immersion freezing based on size and temperature for cloud drops (change 2), as is done in all simulations for immersion freezing of rain. This might affect the solutions of Zhu et al. (2022), who removed the ice number limiter but used a smaller microphysical timestep to control the unrealistic growth of ice number.

3. The new PUMAS code reduces overall LWP and IWP. It is not clear what processes may be responsible for this as it occurs in all PUMAS simulations. The increase in accretion might be partially responsible, as well as the increased sedimentation limiting evaporation. The code can be adjusted to keep more condensed water path by reducing loss rates for liquid and ice, as performed in the 'Tune' experiment.

4. The new PUMAS code reduces the aerosol forcing magnitudes significantly, with offsetting effects on the total ACI. The mechanism may be related to higher ice numbers and less liquid water at high latitudes. There is little net change on aerosol forcing, but reductions in magnitude of SW and LW components.

5. The PUMAS changes have small impacts on cloud feedbacks and climate sensitivity. There are reductions in the N. Hemisphere mid-latitude feedbacks and increases in the S. Hemisphere midlatitudes. Tropical feedbacks increase slightly. Feedback changes appear to be due mainly to the implicit sedimentation (changes 5); reverting this change makes PUMAS look more like the control.

6. The implicit sedimentation calculation (changes 5) also contribute to lowering the aerosol forcing, likely through changes to precipitation production, with less change to LWP. The implicit sedimentation reduces sensitivity to the time step and increases the ice fraction near the freezing level. There does not seem to be an appreciable difference in the time-averaged precipitation or the precipitation process rates due to use of a less accurate implicit method.



Next steps for PUMAS development include integrating the unified ice formulation of Eidhammer et al. (2016), further inves-
420 tigations into ice nucleation and the budget of ice nuclei, and the exploration of flexible form to the size distributions (Morrison
et al., 2020). In addition, the GPU results motivate further work to add GPU directives to all the physical parameterizations in
CAM.

*Code and data availability.* The updated model code described here is available in the PUMAS Repository https://github.com/ESCOMP/
PUMAS/tree/pumas_cam-release_v1.27 as implemented in the Community Atmosphere model https://github.com/ESCOMP/CAM/tree/
cam6_3_075. Derived data sets in CAM and SCAM are available at doi:10.5281/zenodo.7105927.

*Author contributions.* AG performed simulations and wrote the manuscript. TE and KT-C helped develop and integrate the code. HM assisted
with code modifications and editing of the manuscript. JS and JD worked on the GPU code. ZM found the original issues with ice nucleation
and helped analyze solutions with TS. RF assisted with discussions of numerics and implicit sedimentation compared to other models. JZ
helped develop and test some of the changes to ice nucleation.

*Competing interests.* I declare that no competing interests are present.



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
