# Peer review of "Importance of Ice Nucleation and Precipitation on Climate with the Parameterization of Unified Microphysics Across Scales version 1 (PUMASv1)"

_EGUsphere, 2022_

## Author Comment (AC2)

We thank the reviewers for their time and the detailed notes. We have made all of the changes suggested and corrected and clarified as suggested.

We did find one error in our formulation of the use of the Bigg immersion freezing limiter, so this was corrected and the treatment of immersion freezing revised. Instead of shutting off the CNT scheme for immersion freezing, we adopt an approach where we limit the fraction of dust aerosols acting as immersion INP, following what is already used in the ice nucleation code for soot. In addition, we use a more consistent treatment of aerosol number for ice nucleation, as noted in the revised text.

The ice number results are sensitive to these changes, but the conclusions are not. This has been described now in the text, and all the figures updated with new simulations. The conclusions of the paper are not changed, and this is still a minor update to the previous version. New simulations have been run, and updated code and a new archive has been provided (updated doi).

Review #1

This manuscript describes a major development step on the cloud microphysical scheme of the CESM model, including modifications to enable GPU usage with the code. A detailed analysis is performed to test the impact of various changes on the simulations results, including a promising performance test, which demonstrates the potential of GPU usage to improve the speed of high resolution simulations. The code (made available by the authors via github/zenodo) is well structured and properly documented.

The paper fits very well to the scope of GMD and it deserves publications. However, the presentation quality could be improved by addressing the suggestions given below.

- L21-22: *"This frequency bias has been shown to be directly attributable to cloud microphysics (Gettelman et al., 2021)."* Is this a general issue in global modelling or specific to this model? In the cited paper there is no reference to precipitation biases, is this the correct citation?

>> It is a general issue (noted in the previous sentence, adding 'many' for clarity). I apologize, the reference refers to the wrong paper: still Gettelman et al 2021, but the wrong one. It should be 'Machine Learning the Warm Rain Process'. Corrected.

- L35: here I would specify that you are referring to the M3 scheme (although this becomes clear later).

>> Added.

- L110: I understand that the upper limit at 100 cm$_{-3}$ is a free parameter, but could you provide a reason for this choice? Is it supported by observations?

>> We have added further explanation:  At small scales measurements have shown much higher concentrations, up to ~50 cm-3, embedded within broader regions with much lower concentrations < 1 cm-3 (e.g., Hoyle et al. 2005, JAS). Satellite retrievals also show lower concentrations (up to several hundred per liter) over broad regions, e.g. Gryspeerdt et al. 2018, ACP. The main point is that we want to set this upper limit for ice concentration NIMAX to a large value such that it encompasses the range of physical values – in general we want the process parameterizations to control the ice concentration rather than NIMAX, and only impose the limit to ensure values are within a physically reasonable range.

- L224: You may also cite Zelinka et al. (GRL, 2020).

>> Good point. Added.

- L227-L229: Please be more specific on the boundary conditions. What do you mean with averaged GHG and SST? Is this a climatology over a certain period? Are the aerosol emissions 1995-2005 also a climatology or are they applied in a transient manner?

>> Clarified : "Boundary conditions used to force the model are climatological monthly averages over 1995--2000, including greenhouse gases, atmospheric oxidants and emissions of aerosols. The climatological annual cycle is repeated each year."

- L238: *"especially GHGs and SSTs are the same"*, this is confusing, since SSTs are increased in the SST4K simulations.

>> Clarified "(especially GHGs)" only (removed SST here)

- Sect. 4.1: I think this subsection belongs to "Methodology" rather than "Results".

>> Moved as suggested.

- L255: changes #1, #6 and #7 are not mentioned here, but are referred to later.

>> Rephased to list all the modifications going into the PUMAS simulation explicitly

- L264: does this refer to a radius or a diameter? Please specify.

>> Diameter, clarified

- L270: *"high altitude"*, I guess this means p < 600 hPa. Please specify.

>> Specified ("pressures less than 600hPa")

- L271: here and in the following, I would suggest writing the line color and dash in bracket every time you refer to a process, to facilitate following your explanations on the respective figures, e.g. *"accretion of liquid onto snow (dashed-dotted blue line)…"*, etc.

>> Done.

- Table1: it could be helpful adding a Table at the end of section 2, where the processes are briefly summarized, using the same numbering (#1, #2) used here.

>> This is listed in the text in section 2.2 and referred to in the caption. We considered making it a new table there, but it seems to flow better just in the text and then referred to. We do now list everything in the text in section 3 for clarity.

- L275-276: *"and hence an increase in the melting source of liquid"*. Is this the dashed-dotted black line in Fig. 3D-F? Because that does not show an increase.

>> Deleted.

- L284-285: *"by plotting some of the changes separately (not shown)"*. You may consider adding these plots to a Supplement, for the interested readers.

>> We elected not to do this, as the plots do not really elegantly show anything more than is described in the text.

- Fig. 3: the legend is very hard to read, you may consider moving it outside the plot (e.g. on the right part of the figure) and write the processes explicitly instead of using abbreviations, since these are not always clear.

>> **The legend has been moved to outside the plots as suggested, and the process names are now stated explicitly in the legend**

- Fig. 4: please add some horizontal space between the panels, since the label on the x-axis are sometimes overlapping. As for Fig. 3, I would move the legend to the bottom of the figure and enlarge it a bit (e.g., use 3 columns instead of 2). The thicker lines indicating longer timesteps are hard to distinguish, an alternative could be using a lighter color (e.g., blue vs. light blue).

>> We have modified the figure to adjust the legend and make it 3 columns as suggested. We tried using separate colors (different shades) and that actually proved harder to

understand in the figure. It is easier to separate the different sensitivity tests if the different timesteps are all the same color.

- L307-308: Why is the MPACE case less sensitive to time-step than ARM97?

>> Noted some speculation. It's not entirely clear why. It might have to do with the presence of deep convective clouds and deeper clouds which are present throughout the column in ARM97, whereas MPACE has low supercooled clouds and some high clouds.

- L318: you could add that the ice modifications in PUMAS help to reduce the bias w.r.t. EBAF in SW but not in LW.

>> Noted

- Fig. 6A, 6C: could you please comment on the very large bias over the south polar regions? Is that due to the satellite data being less reliable at high latitudes?

>> Yes, the satellite data (based on MODIS clouds) is not reliable at high southern latitudes over the ice sheet. It is clear that the highest water contents are NOT over the S. Pole. Noted.

- Fig. 6: please add a reference for the EBAF dataset. As far as I know, EBAF does not include LWP: which observations are shown for LWP in 6A? Please specify, and if this is not EBAF please correct the legend. Why not including observations also for IWP (6B) and precipitation (6F)? At least for the latter, this should be easy to find (e.g., GPCP, Adler et al. (2018), https://doi.org/10.3390/atmos9040138).

>> Reference for CERES EBAF added. Corrected that the LWP data are from CERES SYN data. IWP data is not compatible with the model outputs since the observations include snow. We have added GPCP precipitation data as suggested.

- Table 3: if possible, it would be interesting to see some values from other models (e.g., the CMIP6 multi-model mean). You could also mention that the total aerosol effect (RESTOM) lies at the lower end of the current estimates (e.g., Bellouin et al. 2020 gives -2.0 to -0.4 W m$_{-2}$ with 90% likelihood).

>> Radiative forcing from different CMIP6 models is limited since there are lots of different configurations used (there are just a few RFMIP experiments that can do it) and they are not exactly the same as this configuration. But noting the Bellouin et al 2020 range is definitely useful. Added.

Minor suggestions:

- Author list: the affiliation #2 comes after #3, #4 and #5. Please rearrange.

>> Done

- L4: the "removing" of the limiter is actually a refactoring, as explained later (L81). Please correct for consistency.

>> Done

- L5: I would write "ice sedimentation" explicitly.

>> It's not just ice, but refers to sedimentation of everything (rain, snow, graupel). Left as is.

- L10: acronym PUMAS not yet defined.

>> Defined now at the beginning of the abstract

- L14: I would add the definition of supercooled liquid water here.

>> Done

- L76: "ontributing" -> "contributing".

>> Whoops. Corrected. Thanks!

- L92: "ice nuclei", the recent literature seems to favour the term "ice nucleating particles". You may consider changing this here and throughout the paper (or use the acronym INPs).

>> Done

- L115: "removed" -> "refactored" (see also comment above).

>> Done

- L145: 1.e-10 -> $10^{-10}$.

>> Changed

- L222: "such as anthropogenic greenhouse gases" -> "such as the increase in anthropogenic greenhouse gases".

>> Changed

- L233: "large" -> "larger".

>> Large scale condensation is the correct term used in the literature.

- L268: "control" -> "Control", for consistency with the naming in Table 1 (this occurs also later).

>> Changed throughout the manuscript when referring to this case

- L303: "cloud and supercooled liquid, as well as cloud ice" -> "cloud (5A) and supercooled (5B) liquid, as well as cloud ice (5C)".

>> Done

- L389, 394, 399 "removal" -> "refactoring" (see also comment above)

>> Done

Review #2

In this study, the authors document scientific and technical updates to cloud parameterizations (and especially ice and mixed-phase clouds) of the CESM climate model. They then assess the impact of those updates on simulated climate, cloud feedback, and aerosol radiative forcing. They find that some updates (removal of an upper limit on ice crystal number and revising to hydrometeor sedimentation) have sizeable impacts on those quantities, which should however be tunable.

The manuscript is well written, very clearly organized. Figures and Tables illustrate the discussion well. The conclusion section is especially efficient at conveying the key messages of the study. The paper is a useful documentation of the PUMAS configuration of CESM, as well as a useful reminder of the power of the detail of physical parameterization at affecting large-scale climate and climate response. The discussion requires clarifications in places, as given in my comments below, but these should only amount to minor revisions.

Comments:

- Line 81: I was at first confused by the apparent conflict between "removing an ice number limiter" in the abstract, and "refactoring of the ice number limiter" on this line. Section 2.2.1 clarifies that one number limiter (NIMAX) has been replaced by another (cap at 100 cm$_{-3}$). It could be worth clarifying throughout.

>> Clarified to refactoring throughout.

- Line 90: On that point, why has NIMAX been removed, instead of updating its calculations (which was incomplete) by adding NICNT and fixing NIMEY? Is that related to the issue discussed in section 2.2.2? How different is NIMAX compared to the new cap value of 100 cm$_{-3}$?

>> The main point is that we want to set this upper limit for ice concentration NIMAX to a large value such that it encompasses the range of physical values – in general we want the

process parameterizations to control the ice concentration rather than NIMAX, and only impose the limit to ensure values are within a physically reasonable range. We have noted this now.

- Lines 120-122: To clarify, has reverting back to Biggs 1953 removed all aerosol-ice interactions, or only some?

>> Only some (immersion freezing only). But Bigg 1953 is now not being used in the final PUMAS code as described above.

- Line 161: The modification of accretion is described as an experiment, rather than a change. It would be good to clarify in the section whether the change was adopted.

>> The change is adopted in the control. Noted.

- Line 263: "scaling the autoconversion of cloud to rain by a further factor of 0.5". Done that mean that autoconversion rates are divided by two in that simulation? I do not understand the "further" in that sentence. Has another scaling been applied elsewhere?

>> Corrected. No, just the additional factor. Removed 'further'

- Lines 267-268: Could note here that graupel changes are briefly discussed later.

>> Noted

- Line 282: Can you quantify what you mean by "modest"? A few percent? Statistically insignificant?

>> Yes, not significantly affecting snow or other hydrometeors. Clarified.

- Lines 286-287 and 305: One downside of implicit sedimentation mentioned in Section 2.2.5 is that it makes the model more diffusive. I suppose the SCAM tests are too short to show an impact there. Is there an indication of the model being more diffusive in the PUMAS-Implicit simulation?

>> There is no indication that more diffusive sedimentation is an issue in either the SCAM or full simulations. Some differences may be seen in the timestep simulations where we discuss this (it's referenced as an issue in section 2.2.5)

- Line 313: Weren't the simulations 6-year long? (Line 227)

>> Yes, corrected. Thanks for catching that.

- Line 363: I wasn't expecting a large change in clear-sky aerosol radiative forcing, since the changes made to PUMAS only directly impact aerosol-cloud interactions. Is there a change in aerosol burden distributions through precipitation change?

>> There is now not a large change in the clear sky forcing.

- Line 383: Are the large differences over the Himalayas in the SW (Figure 10B) and LW (10D) both due to changes in IWP in PUMAS? It would be good to be reminded here of which aerosol-ice interactions are represented in the control and PUMAS simulation.

>> These changes are much smaller and the text has been revised.

Technical changes:

- Line 10: The acronym PUMAS is defined in the title but might need to be defined in the abstract as well.

>> Done

- Line 75: Typo ontributing

>> Corrected

- Line 169: Typo calculation

>> Corrected

- Line 206: Could give a preliminary citation for that manuscript in preparation, as done in the next section for Gettelman, et al 2022?

>> Now submitted, noted.

- Line 233: large -> larger

>> Actually, large is the correct term here.

- Line 302: Parentheses missing around "Figure 5".

>> Corrected (added "in Figure 5")

- Line 319: Need to define the LWP and IWP acronyms here.

>> Done

- Line 324: it's -> its

>> Corrected

- Line 341: integral -> area

>> Corrected

- Caption of Figure 6: Typo radiatve

**>> Corrected**

---

## Author Response (AR2)

Technical Corrections Requested

— Changes: Reference to Fig 7E should be 7D
        • Corrected on line 360
— "Section 3.1 (Sensitivity Tests) should be section 3.2"
It is correct in the manuscript, but there was a mistake automatically
formatting in the 'track changes' version. Sorry about that. Text is
correct.